# X-ray Excited Optical Luminescence of Eu in Diamond Crystals Synthesized at High Pressure High Temperature

**DOI:** 10.3390/ma16020830

**Published:** 2023-01-15

**Authors:** Vasily T. Lebedev, Fedor M. Shakhov, Alexandr Ya. Vul, Arcady A. Zakharov, Vladimir G. Zinoviev, Vera A. Orlova, Eduard V. Fomin

**Affiliations:** 1B.P.Konstantinov Petersburg Nuclear Physics Institute of NRC Kurchatov Institute, 188300 Gatchina, Russia; 2Ioffe Institute, Polytekhnicheskaya Street, 26, 194021 St. Petersburg, Russia; 3V.G.Khlopin Radium Institute, 188300 Gatchina, Russia

**Keywords:** diamond, synthesis, diphthalocyanine, pyrolyzate, lanthanide, X-ray, luminescence

## Abstract

Powder diamonds with integrated europium atoms were synthesized at high pressure (7.7 GPa) and temperature (1800 °C) from a mixture of pentaerythritol with pyrolyzate of diphthalocyanine (C_64_H_32_N_16_Eu) being a special precursor. In diamonds prepared by X-ray fluorescence spectroscopy, we have found a concentration of Eu atoms of 51 ± 5 ppm that is by two orders of magnitude greater than that in natural and synthetic diamonds. X-ray diffraction, SEM, X-ray exited optical luminescence, and Raman and IR spectroscopy have confirmed the formation of high-quality diamond monocrystals containing Eu and a substantial amount of nitrogen (~500 ppm). Numerical simulation has allowed us to determine the energy cost of 5.8 eV needed for the incorporation of a single Eu atom with adjacent vacancy into growing diamond crystal (528 carbons).

## 1. Introduction

The greatly increased interest in the studies and actual applications of lanthanides (Ln) is mainly based on their atomic nature; 4f electrons provide excellent magnetic and luminescent properties which remain relatively stable even when the atoms are ionized, form chemical bonds, and become embedded in crystalline matrices [1]. Most often these elements are used in the form of oxides (LOs), and their physicochemical properties are discussed in a series of recent reviews devoted also to relevant fields of LOs applications [2,3,4,5].

The explicit review [2] summarizes various aspects of scientific activity related to numerous LOs biomedical prospects in theranostics for drug delivery, bio-imaging, cell tracking and labeling, tissue engineering, and cancer treatment. The authors [2] considered the applications of LOs in biosensors and in the contrasting agents for magnetic resonance imaging (MRI), in the preparations reducing oxidative stresses, and in the provision of antimicrobial effects.

On the other hand, for a wide industrial implementation of Los, it is necessary to design the biosensors and electronic devices for electrochemistry and for the control of chemical and physical parameters (temperature, humidity, gas impurities) [3]. The industrial applications of LOs are relevant in corrosion protection, catalytic and photoactive reactions, and in solving ecological problems [4].

Many more such opportunities could be realized by means of Ln incorporation into host materials, e.g., to improve the performance of photovoltaic cells and to construct portable energy devices, dosimeters, and shielding glasses against radiation [5].

Among matrices which can be doped with Ln for biomedical applications, the chemically inert, biocompatible, radiation resistant, transparent, and luminescent diamonds are most profitable. However, embedding large-sized Ln atoms into densely packed diamond structures remains a complicated problem. The Ln intercalation into the diamond lattice is relevant in fundamental and practical aspects and stimulates a search for ways of doping diamonds for various purposes; this is especially the case in optoelectronics and biomedicine owing to the pronounced luminescent properties of embedded lanthanides when excited by UV, visible light, X-rays [6,7,8]. Such problem solving will enable the production of unique diamond crystals which are unknown in nature.

Meanwhile, in tested fossil diamonds, the neutron activation analysis showed extremely low concentrations of lanthanides (~10^−4^ wt.%) localized in phosphate inclusions with a density close to that of diamond [9]. The authors [6] remarked that usually the ion implantation to dope crystal with external elements irreversibly damages the host matrix. This problem complicates the creation of centers of luminescence in diamonds due to their graphitization in the annealing process undertaken to restore the diamond structure.

Therefore, the authors [6] developed an alternative method to incorporate europium into diamonds. They oxidized the surface of crystalline seeds (5 nm), deposited on them a polyelectrolyte with linked chelate molecules containing Eu (III) and then condensed carbon from the gas phase on the samples (CVD method) [6]. Even so, the concentrations of Eu ions in the grown diamonds were too low (~10^−4^ wt.%). The density functional modeling made it possible to simulate defect formation such as that cause by the Eu atom in different charge states (3+ mainly) with attached 1–3 vacancies. The configuration with one vacancy turned out to be the most stable. In this case, a coordination sphere of six carbon atoms was formed around the Eu atom, and the authors detected the electron transitions in Eu (III) involving f-orbitals [6].

To introduce Ln atoms into microcrystalline diamonds, detonation nanodiamond particles (~5 nm in size) have been modified with Gd or Eu ions grafted to the diamond surface through ion exchange with carboxyl groups. Then the samples were sintered at high pressures and temperatures (7 GPa, 1300–1500 °C, HPHT method) [8]. In these experiments, hydrocarbons and alcohols served instead of metal catalysts. In following energy-dispersive X-ray spectroscopy tests on the synthesized diamonds, the upper estimate of Ln amounts in diamonds was found of ~0.01 at.%, while at the limit of the accuracy of measurements.

The authors [10] synthesized diamonds in the presence of Ln metallic inserts in the reactor (HPHT), but did not observe in the crystals any luminescence centers with Ln atoms. In addition to CVD and HPHT methods, the Ln embedding into diamonds has been performed by heat treatment (450; 700; 1000 °C) of diamond powders impregnated with water–alcohol solutions of Eu(NO_3_)_3_·H_2_O [11]. In the samples, the authors detected the enhanced excitation of Eu^3+^ ions by UV radiation (280 nm) comparative to europium salt. This was explained by the appearance of Eu-O-C bonds through carboxyl groups at the diamond surface. However, in this case, there are no reasons to expect the intercalation of Ln atoms into the diamond lattice.

The cited works are relevant for theranostics, which trends to use various nanostructures (quantum dots, metal or oxide particles, etc.) promising new functional abilities, especially via X-ray activation. Meanwhile, even a detailed review [12] did not report on the application of diamonds in this area. Mainly, this is explained by the substantial difficulties of doping them with Ln and other metals. To date, no satisfactory solution has been found for the problem of introducing Ln into the diamond lattice. Note, heavy atom intercalation into diamond needs substantial energies (14–17 eV for Eu^3+^ coupled with 1–3 vacancies in a crystal fragment of 64 atoms) [6] because of the dense packing and strong interaction of carbons in diamond. Highly fluorescent lanthanides embedding into diamond lattice can be facilitated by the effect of atoms’ contraction with the increase of atomic number [13]. For instance, Tb has an atomic radius (*r*_Tb_ = 0.175 nm) of ~5% less than that for Eu (*r*_Eu_ = 0.185 nm) when the atomic number is increased by ~3% (*Z*_Eu_ = 63, *Z*_Tb_ = 65). The difference *r*_Eu_ − *r*_Tb_ = 0.010 nm is significant because it achieves ~14% of the carbon atom radius (0.070 nm) [13]. Therefore, to vary diamond doping, it is interesting to use a series of Ln atoms with different atomic numbers.

The aims of our work included the development of new methods for doping diamonds with lanthanides (by the example of europium) to achieve concentrations of heavy atoms by an order of magnitude or greater than the values achieved in synthetic or natural crystals. We planned to design and use a special precursor with nanosized metal–carbon particles capable of transforming into diamond structures with europium atoms incorporated into the lattice at high pressures and temperatures. These developments serve developing technologies in the production of transparent micro(nano)diamonds with strong luminescent and magnetic properties for applications in biomedicine. The expected results imply the use of doped diamonds as markers, contrast agents in magnetic resonance imaging, optically active platforms for targeted delivery of photosensitizers in photodynamic therapy (PDT). Along with this, for the progress in the X-ray photodynamic therapy (X-PDT), it is urgent to use such diamonds as converters of X-ray radiation into the optical range for excitation of a photosensitizer attached to the diamond in order to generate the most reactive singlet oxygen in the foci of the disease and effectively destroy tumors.

First, we started the studies with a highly luminescent Eu element and searched for effective precursors, in which one or several Eu atoms could be isolated inside durable carbon shells with a free volume suitable for the rearrangement of metal–carbon structures into the diamond lattice under high pressures and temperatures (HPHT). Obviously, such objects as endometallofullerenes (EMF) could serve as promising candidates to be transformed into diamonds upon the compression of molecular carbon shells around metal atoms. However, such experiments are still unknown due to very low availability of EMFs even in laboratory quantities.

As an alternative to EMF, we considered diphthalocyanines (EuPc_2_). During pyrolysis in Ar atmosphere (900 °C), such molecules, composed of two planar ligands connected by metal atom, lost mainly light elements (H, N). As a result, in a molecule, the ligands have free bonds: those linking causes metal atom closing inside a carbon shell (~1 nm in size) [14]. We also imposed an encapsulation of a few metal atoms in a shell by combining several molecules into globules forming a porous matrix where the metal atoms are firmly kept and are released only by heating above 1200 °C when amorphous carbon transformes into graphite [14,15]. To convert such Eu-containing pyrolyzates into diamonds, we have applied the HPHT method using a catalytic additive (pentaerythritol). First, we sought to find conditions for the formation of Eu-doped diamonds and to synthesize metal-enriched crystals. In the following experiments, we aimed to determine the Eu content in diamonds and study their structure and physicochemical properties along with the simulation of the growth of crystals with the incorporation of Eu into the diamond lattice.

## 2. Experimental Section

### 2.1. Materials

We have synthesized doped diamonds by using special precursors (pyrolyzates) being the carbon matrices encapsulated europium atoms with the atomic proportion Eu:C = 1:(30–40). We have prepared the pyrolyzates from the diphthalocyanines (EuPc_2_) [14,16] in which molecules the planar C_32_H_16_N_8_ ligands (Pc) are firmly linked through a metal atom (M = Eu) (Figure 1).

First, for the synthesis of LnPc_2_ molecules, the organic component, o-phthalonitrile, was placed into a quartz reactor and washed with argon flow [14]. At a constant flow of argon (~0.5 L/min), the temperature in the reactor was raised to 220–250 °C. Then the salt (Eu acetate) was added in a mass ratio of 1:6 to the melt of o-phthalonitrile by stirring the mixture. The reaction between the components (25–30 min.) led to the formation of EuPc_2_ compound (Figure 1). After EuPc_2_ synthesis completion, the temperature was increased to 350–400 °C to distill off the excess o-phthalonitrile and other reaction by-products condensed in the top of the reactor. At the second stage, to produce the pyrolyzate from the EuPc_2_ molecules, we have increased the temperature in the reactor to 850–900 °C. In this process, the destruction of EuPc_2_ molecules led to the formation of target product being finely dispersed carbon–metal powder. Previously, these substances were characterized by neutron scattering; their fine structure was determined by a large volume fraction of pores (~50%) having a wide distribution in size (10^0^ –10^2^ nm) [17,18]. At larger scales, the irregular “labyrinth structures” have been detected by atomic force microscopy (AFM) on the surface of porous pyrolyzate particles formed of carbon aggregates (Figure 2).

In following X-ray wide angle scattering experiments (XRD—X-ray diffraction), the pyrolyzate powder showed the completely amorphous structure discussed below. According to previous data [14], such carbon matrices strongly retain metal atoms. The immobilization of heavy elements inside pyrolyzates was tested by heating up to 1600 °C [14]. In following diamond synthesis to achieve a catalytic effect, we used this temperature resistant precursor filled with pentaerythritol, C(CH_2_OH)_4_.

### 2.2. Synthesis and Purification

To produce Eu-doped diamonds, we applied the HPHT method optimized in the synthesis of microdiamond powders from different carbon precursors such as detonation nanodiamond [19], shock-wave polycrystalline diamonds [20], and carbon black [21] together with pentaerythritol or ethanol as the hydrocarbon sources.

The HPHT method [22] is the most economical and allows the production of large diamond crystals by modifying them with various elements, setting the composition of the precursor and synthesis conditions in terms of pressure, temperature, process duration, and when the diamond as a carbon phase is stable, in contrast to the situation when synthesis proceeds from the gas phase (chemical vapor deposition, CVD) at low pressure, i.e., under metastable conditions for diamond. Depending on the scientific and technological tasks, various catalysts (transition metals, nanocarbon, hydrocarbons) can be introduced into the precursor to enhance the efficiency of synthesis and product quality [23,24].

Thus, it is possible to preset the size of the resulting crystals, control their nitrogen concentration, thermal conductivity, optical transparency, electrical resistivity, and hardness, and ensure high perfection of the crystal structure. The progress in technologies for the industrial production of high-quality diamonds, including jewelry ones, is most associated with the HPHT method. The disadvantages of the method include the somewhat unreliable controlled reproducibility of the results and the possible ingress of impurities into the diamonds from the material of the chamber. Along with this, in the HPHT synthesis conducted above the graphite–diamond boundary line, an uncertainty in line position may lead to some defects during diamond crystallization from the graphite phase (plane incompleteness, inclusions). A detailed comparison of the methods for the diamond synthesis is given in reviews [24,25].

In practice, depending on the task, the size and quality of powdered diamonds can be chosen. Then, in accordance with the requirements for diamond powder, the method and conditions of synthesis should be selected. We tried to obtain nanosized and submicron powders doped with europium for use as luminescent markers. To achieve this aim, there are two possible methods: a detonation process, or a synthesis at high pressure and temperature. So far, no successful doping attempts in the detonation process are known. However, this is attainable via the HPHT method.

It is necessary to choose such conditions and synthesis precursors so that nanosized particles are immediately formed [26,27,28] or micron-sized diamond powders are synthesized, which can be crushed [29]. Usually, a longer synthesis process enables the growth of larger and better crystals. Presently there is a problem of obtaining submicron powders. Methods for diamond crushing are considered in [29].

In our case, a short process (~10 s) is ideal in terms of both getting satisfactory quality crystals and shortening synthesis time. The toroid-type high-pressure chamber used consists of minimal of elements and does not require a long preparation for the synthesis of small-sized diamonds. This chamber is not intended for the production of large crystals. By means of the HPTP method with the toroid-type chamber, we achieved the aims of producing small diamonds enriched with europium ions Eu^3+^. Such crystal types should be more suitable for biomedical applications than the particles of rare earth oxides with some challenges concerning toxicity risks [2].

In the experiments, the pyrolyzate powders mixed with pentaerythritol (50:50 wt.%) and packed into toroidal containers with graphite bushings were exposed to 11 s at high pressures and temperatures (7.5–8 GPa, 1800 °C). The sintered samples (cylinders) have been milled to produce fine powders then mixed to prepare a raw sample SEu1 containing diamond and other phases (graphite, aragonite, Eu-hydroxides) (Table 1). The following procedures included etching the samples (SEu2) in hydrochloric acid followed by the hydrostatic separation of graphite and diamond in bromoform, CHBr_3_ (SEu3, Table 1).

### 2.3. Methods

To study the powder samples, we have used the X-ray phase analysis (Rigaku Smart Lab III diffractometer, copper anode, Bragg-Brentano geometry, accelerating voltage 40 kV, current 30 A, Soller slits 2.5°, recording step 0.01°, speed 5°/min) and determined the composition of the crystalline phases by using the ICSD PDF2 database.

To determine the Eu concentration in diamonds, we applied radiometric X-ray analysis (XRA) [30], which detects extremely small amounts (several ppm) of impurities in materials. This method, also known as XRF (X-ray fluorescence), is based on the measurement of secondary fluorescent X-rays emitted by a sample when it is excited by an X-ray source [31].

In the analysis of the samples with various Eu-contents, we exploited different sources (i.e., XRF@109Cd, XRF@241Am) with moderate or high power (^109^Cd radionuclide, activity of 0.36 GBq, quanta energy 22–25 keV, set of lines; ^241^Am, activity of 80 GBq. quanta energy 59.5 keV). As the references, the Eu (III) oxides served when dissolved in water with 1 M HNO_3_.

The analysis has been performed on the X-ray spectrometer consisting of PGT1000-13 (GmbH) detector with the energy resolution of 200 eV for Kα radiation of Fe (6.4 keV) and ORTEC spectrometric system. The dependences of the radiation intensity *I(E)* on the energy of photons from probes (reference solutions, SEu1,2,3) were recorded, which showed spectra with Kα1, Kα2 lines for Eu atoms, the concentrations of which were estimated by the method [32].

Since the aim of the synthesis was to produce the luminescent diamonds due to Eu atoms in crystals, we tested them by the method of X-ray exited optical luminescence (XEOL) [33,34]. We studied the samples by irradiating them with X-rays (quantum energy 8 keV, wavelength λ = 0.154 nm) and detecting UV and visible radiation with the wavelengths λ = 380–900 nm. Through the system of slits, the radiation from the X-ray tube (1.5BSV29-Cu) entered the crystal monochromator separated quanta with the energy of 8.0 keV from a full spectrum. The slits and Soller collimator formed the X-ray beam 2 × 2 mm^2^ (intensity of 1.5 × 10^6^ s^−1^) directed to the sample surface. The AvaSpec ULS2048L optical spectrometer (range of 380–900 nm) provided a registration of luminescence photons induced by X-rays in the sample (finely dispersed powder) poured into a flat container, one of the walls of which was made of Mylar. For physicochemical analysis of samples, we used standard FTIR, Raman spectroscopy and SEM.

In all XRD, XRF, XEOL, and FTIR experiments performed at 20 °C, we used powder samples (mass of 100 mg). The diamond crystals were micrometer sized.

### 2.4. Simulation

At the final stage of the work, we adapted computer methods for modeling the growth of perfect nanocrystals and crystals with one embedded Eu atom (MM2 force field method, PerkinElmer Chem Office, Chem3D module) [35,36].

## 3. Results and Discussions

### 3.1. Structure of Pyrolyzate

Before the diamond synthesis, we analyzed pyrolyzate structure by XRD (20 °C); it showed no crystalline reflects in the angular range 2*θ* = 10–133 deg. For amorphous powder irradiated with X-rays (wavelength λ = 0.154 nm), the scattering intensity *I*(*q*) vs. scattering vector modulus *q* = (4π/λ)sin (*θ*) exhibited only broad peaks at *q*_1_~50 nm^−1^, *q*_2_~30 nm^−1^, and *q*_3_~17 nm^−1^ (Figure 3). The first peak revealed the correlations at the distance *L*_1_~2π/*q*_1_~0.13 nm, comparable to the bond length between carbon atoms, the second and third peaks displayed europium–carbon and other interatomic correlations with the lengths *L*_2_~2π/*q*_2_~0.2 nm and *L*_3_~2π/*q*_3_~0.4 nm.

To decode pyrolyzate structure, we restored the spectrum of spatial correlations *G*(*R*) from the data by indirect Fourier transform (ATSAS package) [37,38,39] (Figure 4). The profile of *G(R)* = *R*^2^γ(*R*) is defined by the pair correlation function γ(*R*) for scattering centers (carbons, europium atoms, inhomogeneities in atomic packing) at the distances *R* = 0–1.3 nm. The *G*(*R*) distribution represents a wide peak with maximum position indicating the radii of detected metal–carbon particles, *R*~0.5–0.7 nm, exceeding the size of EuPc_2_ molecule. Thus, as a result of pyrolysis, the molecules are united into globular particles and each of them captures several Eu atoms in a carbon shell.

The core-shell model allowed us to describe the data (Figure 4) for the spectrum *G*(*R*) as being a superposition of partial correlators for electron density within individual carbon atoms or Eu clusters (*G*_1_, *G*_2_), the correlations between Eu clusters and neighboring carbons (*G*_3_), and atom pair correlations in the shell (*G*_4_):*G*(*R*) = ∑*g*_i_*G*_i_, *i* = 1,.4; *G*_1,2_ = *R*^2^·exp[−*R*^2^/*r*_1,2_^2^], *G*_3,4_ = *R*^2^·exp[−(*R* − *R*_3,4_)^2^/*r*_3,4_^2^].(1)
Here, we neglected more extended cross-correlations of Eu atoms with carbon layers since the function (1) provided a satisfactory fit (Figure 4) with the coefficients g_i_ (amplitude factors). The other fitting parameters are the correlation lengths *r*_1,2_ and *R*_3,4_ with the dispersions *r*_3,4_ (Table 2). In total, for globular structure described by the spectrum in Figure 4, we have found also the gyration radius *R*_GP_ = 0.48 ± 0.01 nm, which is close to the correlation length *R*_4_. These lengths characterize a core-shell particle (Figure 5).

The parameters *r*_1,2_ = (4/3)^1/2^*r*_g1,2_ define the gyration radii *r*_g1,2_ for electron density in carbon atoms and Eu clusters. At a longer scale, the *R*_3_ is a characteristic distance between the Eu atoms in the cluster and carbons in the layer adjacent to the cluster surface. The *R*_4_ with the dispersion *r*_4_ is the most probable length for the correlations between carbons in the entire shell around the Eu cluster.

The *r*_1_ gives the gyration radius of atomic electron shell *r*_g1_ = (3/4)^1/2^*r*_1_ = 0.056 ± 0.002 nm, and the spherical approximation defines the atomic geometric radius, *r*_C_ = *r*_g1_(5/3)^1/2^ = 0.073 ± 0.003 nm, corresponding to carbons (*r*_C_ = 0.070 nm) [13]. Similarly for Eu clusters, the *r*_2_ determines their gyration and geometric radii, *r*_g2_ = 0.21 ± 0.03 nm, *r*_CL_ = *r*_g2_(5/3)^1/2^ = 0.27 ± 0.03 nm. The radius *r*_CL_ is by 35% greater than that for single Eu atom, *r*_Eu_ = 0.185 nm [13]. Hence, a cluster integrates a number of Eu atoms ν ≈ (*r*_CL_/*r*_Eu_)^3^ ≈ 3. On average a globule which captured Eu cluster is composed of ν ≈3 diphthalocyanines linked via free bonds of molecular ligands retained carbons but lost H, N atoms by pyrolysis.

Such globules are aggregated into a metal–carbon matrix which has a brutto formula EuC_X_ (X = 30–40) with residual nitrogen [14]. At the parameter ν ≈ 3, a globule with Eu atoms has a number of carbons, *n*_t_ ≈ 100.

Inside a massive carbon shell, the Eu cluster with the radius *r*_CL_~0.3 nm is coordinated at characteristic distance *R*_3_~0.4 nm with adjacent carbon layer of thickness δ_1_ = 2*r*_3_/√2 = 0.16 ± 0.04 nm corresponding to carbon atom size. Cluster-shell contacts lead to complex formation with charge transfer from Eu to carbon atoms. This is confirmed in pyrolyzate powders by gamma-resonance spectroscopy revealing mostly the Eu^3+^ ions [40].

Longer pair atomic correlations in a globule consisting mainly of carbons are characterized by the distance *R*_4_ with the dispersion *r*_4_ being a measure of carbon shell thickness δ_t_ = 2*r*_4_/√2 = 0.56 ± 0.03 nm (~4 layers). In total, for such a globular structure described by the *G*(*R*) spectrum (Figure 4), the gyration radius *R*_GP_ = 0.48 ± 0.01 nm is quite close to the correlation length R_4_.

We obtained the extended structural information by the integration of correlators. The integral *s*_1_ = ∫*G*_1_*dR*~*N*_C_*Z*_C_^2^ is proportional to the quantity of carbons (*N*_C_) in the sample, and their squared atomic number (*Z*_C_ = 6). For Eu clusters, the integral *s*_2_ = ∫*G*_2_*dR*~ν*N*_Eu_*Z*_Eu_^2^ is defined by the quantity of Eu atoms (*N*_Eu_), squared atomic (*Z*_Eu_ = 63), and aggregation (ν) numbers. The data on a cluster contact with neighboring carbons we obtained from the integral *s*_3_ = ∫*G*_3_*dR*~2*n*_1_*N*_Eu_*Z*_Eu_*Z*_C_. Finally, the integral for the whole carbon shell, *s*_4_ = ∫*G*_4_*dR*~*n*_t_*N*_C_*Z*_C_^2^, gives the number of constituent atoms (*n*_t_)

Combining the equations (ν/*n*_t_) = *N*_Eu_/*N*_C_, *s*_4_/*s*_1_ = *n*_t_, *s*_4_/*s*_2_ = (*N*_C_/*N*_Eu_)^2^(*Z*_C_/*Z*_Eu_)^2^, *s*_3_/*s*_1_ = 2*n*_1_(*N*_Eu_/*N*_C_)(*Z*_Eu_/*Z*_C_), we have found all the parameters: the number of carbons in a globule, *n*_t_ = 88 ± 23; the atomic C:Eu proportion, *N*_C_/*N*_Eu_ = 33 ± 8; the Eu aggregation number, ν = (*N*_Eu_/*N*_C_)n_t_ = 2.7 ± 0.7; the number of carbons near central Eu cluster, *n*_1_ = 15 ± 7. The parameters ν and *n*_t_ are quite close to the values obtained from the radii *r*_1,2,3_, and atomic proportion Eu:C = 1:(30–40) according to established stoichiometry [14].

In previous studies [14,15,16,17,18], similar pyrolyzates used for the immobilization of heavy nuclides have exhibited an excellent chemical and structural stability by heating up to the temperatures 1200–1600 °C. A substantial release of Ln atoms from such materials has been detected only above 1200 °C [14]. For Eu atom migration, we estimated the activation energy E_AEu_~2.3 eV, which turned out to be three times lower than the energy of vacancy formation in graphite [41]. Hence, this barrier is related to the diffusion of atoms from the pores which become opened in the matrix upon heating.

This heat-resistant precursor retaining heavy atoms can serve as a precursor for doping diamonds in the HPHT process. To intensify this process, we have combined the precursor with a hydrocarbon substance (pentaerythritol), which revealed the catalytic properties in the synthesis of diamonds from carbon clusters of 40 nm in size [21]. We have saturated the pyrolyzate with pentaerythritol (Figure 6) which penetrates into fine pores of the carbon matrix where the interaction of molecules with Eu atoms is made possible by their release from the carbon material during HPHT.

In this way, we realized a favorable regime of transformation of pentaerythritol carbons into diamond with Eu-inclusions when the substance in pores is compressed at high temperatures. In the HPHT process, the Eu atoms firmly kept in globules of pyrolyzate became integrated into diamonds growing from the carbon matrix. Both mechanisms promote in diamond formation with Eu that is discussed below.

### 3.2. X-ray Diffraction and Phase Analysis

XRD patterns for the samples SEu1 and SEu3 before purification and in their final, highly enriched state are shown in Figure 7. As we found, the raw sample SEu1 consists of graphite, diamond, aragonite (CaCO_3_), and europium hydroxide, Eu(OH)_3_. For these phases, the mass contents are given in Figure 7. Aragonite contamination is caused by a partial incorporation of high-pressure chamber material into sintered sample. Some amount of graphite came from the graphite heater, the other part was formed as a result of crystallization of the carbon belonging to pyrolyzate and pentaerythritol under synthesis conditions.

As will be seen from the FTIR spectra, the diamond surface is covered with hydrogen. Therefore, it can be assumed with a high probability that the synthesis occurs in a hydrogen-reducing medium. This suggests that under the synthesis conditions, europium was in the form of hydride, EuH_2_, and turned into europium hydroxide, Eu(OH)_3_, at room conditions after the removal of temperature and pressure. 

Etching the SEu1 in hydrochloric acid resulted in the removal of aragonite and Eu hydroxide. A significant part of graphite was removed by the hydrostatic separation of diamond from graphite in bromoform. As a result, we obtained the sample SEu2 with the proportion diamond:graphite = 52:48 wt.% (Table 1). Finally, well purified sample SEu3, which contained 95 wt.% of diamond and 5 wt.% of graphite. In diffraction, except of graphite crystals, we detected also a nanographite fraction in the range of 2*θ* = 17–38 deg. (wide halo, Figure 7).

### 3.3. XRF, XEOL, TEM, and Raman Results

Primarily, in the raw sample Seu1, the XRF analysis showed a high Eu fraction (5.7 wt.%, Table 1). According to the calculations, in the SEu1, only 3.9 wt.% of europium has entered the hydroxide phase. A residual Eu part (1.8 wt.%) has retained in diamond and graphite. In the sample SEu1, the Eu atoms are observed in the form of Ln^3+^ ions; this is evident from the XEOL data (Figure 8) where the peaks of Eu^3+^ ions luminescence are visible at wavelengths λ~592; 616; 696 nm against a wide emission band (400–700 nm) arisen from the components substantially contributing to the total intensity (diamond, graphite, aragonite) [42,43,44,45]. The characteristic Eu^3+^ emission bands should be attributed to the electron transitions 5D_0_ → ^7^F_1,2,4_ [46,47,48]. At the same time, the spectrum in Figure 8 did not show characteristic radiation from the ions Eu^2+^ (~500 nm) [49] which may present only in minor quantities.

The data in Figure 8 show intense luminescence peaks, predominantly from Eu in the hydroxide phase. The emission features for Eu ions are mainly preset by the atomic environment of ions that is determined by the chemical structure of the molecules and the crystal structure of the hydroxide. This predetermines the charge state of Eu and the positions of the electronic levels responsible for the optical emission. According to the following XRF experiments, the Eu content in diamonds and graphite is two orders of magnitude lower than in the hydroxide phase. Therefore, Eu emission from carbon phases does not make a significant contribution to the luminescence, and the difference between the atomic environment of Eu ions in carbon phases and the atomic coordination of Eu in hydroxide has practically no effect on the total spectral pattern for the sample SEu1 (Figure 8).

Generally, in Ln atoms, the positions of the energy levels of 4f-electrons are weakly dependent on the changes in the crystal field of the environment due to effective screening by the outer 5s and 5p orbitals. As a result, the environment of Ln does not strongly affect the intensity of transitions. For instance, the peak related to the ^5^D_0_–^7^F_1_ magnetic dipole transition is not sensitive to point symmetry of Eu^3+^ ions locations because it is parity allowed [50,51]. However, some transitions are hypersensitive to the changes in the Ln coordination sphere. The analysis of spectroscopic data on hypersensitive transitions is a useful tool in the study of Ln. In the case of Eu, there is a supersensitive transition ^5^D_0_ → ^7^F_2_. The authors [50] studied characteristic optical properties of Eu^3+^ ions in the cubic Y_2_O_3_ host structure and detected the emission spectrum which is composed of ^5^D_0_ → ^7^F_j_ (j = 0, 1, 2, 3, 4) transition lines of Eu^3+^ with the ^5^D_0_ → ^7^F_2_ hypersensitive transition (611 nm) being the most prominent emission due to the Eu^3+^ ions in the lattice not being in centrosymmetric positions.

In the other sample’s Y_3_Al_5_O_12_: Eu (YAG), most of Eu emission is concentrated in the 590 nm orange line from the ^5^D_0_ → ^7^F_1_ transition of Eu^3+^ [51]. The 610 nm red line from the ^5^D_0_ → ^7^F_2_ transition appears but is relatively weak. The ions Eu^3+^ enter the garnet structure on the 8-coordinated Y^3+^ site with site symmetry D_2_. As a result, the luminescent intensity is concentrated mainly in the magnetic dipole transition (590 nm) rather than the forced electric dipole transition (610 nm). The results [50,51] illustrate a significance of central symmetry in the coordination of Eu^3+^ ions in the host lattice. The ratio of integral intensities of luminescence, R_21_ = I(^5^D_0_ → ^7^F_2_)/(^5^D_0_ → ^7^F_1_) can serve as a measure of central symmetry violence. The greater the ratio, the less symmetrical the environment of the europium ion in matrix. In the review [52], the electronic states, energy levels, and transition intensities are discussed regarding to the electric dipole nature of 4f electron transitions and the degree of centrosymmetry of europium site.

At the next stage of the experiments, we have purified the sample SEu1. As a result, the obtained powder sample SEu2 was substantially enriched with diamonds and has shown the amount of Eu, *C*_Eu_ = 88 ± 5 ppm (Table 1), determined in the XRF@241Am tests.

Finally, we have achieved a maximal segregation for diamonds, diamond:graphite = 95:5 wt.%, in the sample SEu3 with Eu content of 55 ± 5 ppm (Figure 7, Table 1). The purified diamonds demonstrated a high quality judged by SEM and showed good submicron(micron)-sized crystals with the smooth facets and a sharp cut (Figure 9).

The Raman spectrum for these diamonds possessed a characteristic peak (Figure 10) which obeys lorentzian *L*(κ)~[(κ − κ_max_)^2^ + Γ^2^]^−1^ with a wavenumber at profile maximum κ_max_ = 1331.1 ± 0.1 cm^−1^ and a small width Γ = 2.19 ± 0.05 cm^−1^, approaching the experimental resolution limit. That this contracted peak in the position is close to the one for bulk diamond (1332.5 cm^−1^) [53,54] indicated that the monocrystals (>100 nm) were in agreement with the SEM pattern (Figure 9) and XRD showed narrow diamond reflexes (Figure 7).

By XRF@241Am, we detected a characteristic X-ray fluorescence from Eu atoms in the powder sample SEu3 compared to the reference solution with Eu(III) ions (Figure 11). Both probes showed emission intensities *I(E)* depending on photon energy with the peaks at *E*_m1_ = 40.9 keV and *E*_m2_ = 41.6 keV corresponding to Eu Kα2 and Eu Kα1 lines (Figure 11).

Following data processing with background subtraction and comparisons of the integrals over emission peaks for the SEu3 and reference sample allowed us to find the Eu content in the SEu3 sample, *C*_Eu_ = 55 ± 5 ppm. The Eu atomic fraction *C*_EuAt_ = 4.2 × 10^−4^% matches to an Eu atom per 2.3 × 10^5^ carbons (crystal fragment~20 nm in size). The presence of Eu^3+^ ions in the sample was confirmed by XEOL (Figure 12).

The luminescence spectrum (Figure 12) demonstrates a high emission in a wide band (450–600 nm) which is usually observed in diamonds due to electron-hole pairs generation by X-ray absorption and subsequent pairs recombination on various lattice defects, including impurities and dislocations [42,43]. On the spectrum tail, we fitted the Eu specific peaks with lorentzians, *L*(κ) = A_0_[(κ − κ_max_)^2^ + Γ^2^]^−1^ and found their amplitudes (A_0_), wavenumbers at the maxima (κ_max_), and linewidths (Γ). All the lines at λ_max_ = 1/κ_max_ = 602 ± 1; 633 ± 1; 690 ± 1 nm have small widths, Γ/κ_max_ = 1.8 ± 0.1; 2.0 ± 0.1; 1.0 ± 0.1%.

The fitting parameters have defined the maximal peak intensities, A_0_/κ_max_^2^, in the proportion 0.6:1:0.3 (Figure 12), which is not so far from the relationship between the intensities of Eu peaks in Figure 8 for the raw sample SEu1. Both spectra (Figure 8 and Figure 12) represent the luminescence for the transitions 5D_0_ →^7^F_1,2,4_ [46,47,48], although maxima positions somewhat differ in these cases. In the highly purified SEu3 sample, the shifted and broadened peaks (Figure 12) relative to the data for raw sample SEu1 (Figure 8) indicate a significant difference in the atomic environment of the Eu ions localized in the diamond lattice or intercalated in graphite as compared to the phase Eu(OH)_3_ in the sample SEu1. In graphite, the Eu atoms localization between atomic planes is expected, in as far as the diamond lattice, it is a preferable formation of Eu^3+^ complexes with 1–3 vacancies when a heavy atom is coordinated with six carbon atoms [6]. In this case, lattice local deformations and stresses are inevitable depending on the number of vacancies around the Eu^3+^ ion; this is also a factor leading to a shift and broadening of the Eu^3+^ luminescence bands (Figure 12).

At the same time, the influence of the aforesaid factors on the luminescence intensity for various bands is very different. As we discussed above, it is reasonable to calculate the coefficient R_21_ = I(^5^D_0_ → ^7^F_2_)/(^5^D_0_ → ^7^F_1_) in order to estimate a degree of centrosymmerty for the Eu sites in the samples. From the data in Figure 8 and Figure 12, we have evaluated the coefficients R_21H_ ≈ 2.8 and R_21D_ ≈ 2.0, respectively. A relationship between the coefficients, R_21H_ > R_21D,_ testifies a stronger violence of centrosymmetry for Eu^3+^ ions in the lattice of Eu(OH)_3_ compared to their position in the diamond lattice and in the residual graphite fragments. In the sample SEu3 the Eu sites in total turned out in more symmetric atomic surrounding than that in Eu(OH)_3_ phase.

Probably, this peculiarity should be attributed to varied numbers of vacancies (1–3) near Eu^3+^ ions [6]

The evaluated spectral characteristics in Figure 12 enabled us to find the ratio of summary intensity integrals over the Eu peaks to the integral over the band, *I*_P_/*I*_B_ ≈ 2.5%. Surprisingly, even at extremely low quantity in the sample, the Eu atoms are explicitly visualized in the emission due to X-ray high absorption by heavy atoms with atomic number (*Z*_Eu_ = 63) exceeding by an order in magnitude the number for carbons (*Z*_C_ = 6).

The linear absorption coefficient μ~ρ*Z*^4^/*AE*^3^ for X-rays in a substance is proportional to its density (ρ). It strongly increases with the atomic number (*Z*) of constituent atoms and decreases with atomic weight (*A*) and quanta energy (*E*) [33].

In the sample SEu3 with doping degree C_Eu_ = 55 ppm, the increase in the absorption coefficient is Δμ/μ = (*C*_Eu_*A*_C_/*A*_Eu_)(*Z*_Eu_/*Z*_C_)^4^ ≈ 5%, where *A*_C_ and *A*_Eu_ are atomic weights for carbon and Eu, respectively. The ratio *I*_P_/*I*_B_ ≈ 2.5% is half as much as Δμ/μ due to nonradiative dissipation of absorbed energy.

The XRF and XEOL methods are in reasonable agreement and confirm the presence of Eu^3+^ ions in the highly diamond-enriched sample SEu3. On the other hand, in the case of XEOL, one can suspect also other channels of luminescence generation due to various lattice defects attributed also to the impurities (N, O, H).

The FTIR spectrum (Figure 13) for the SEu3 showed different nitrogen impurities: neutral atoms at lattice sites (N^0^ or P1 defects), pairs of atoms in neighboring sites (2N, A-aggregates), and ions N^+^ (C^+^ defects) (Table 3).

Among them, A-aggregates are most common, and C+ defects are least of all present. In addition, the diamond surface is saturated with hydrogen. Despite substantial amounts of nitrogen (471 ppm), the crystals possess a good quality according to the data of SEM and Raman spectroscopy (Figure 9 and Figure 10).

Detected by FTIR, the defects (Table 3) are practically invisible in the XEOL experiments because such large amounts of nitrogen (Table 3) are not appropriate for X-ray excitation of luminescence. Numerous defects preferably absorb in optical diapason and this effect is used to detect nitrogen in diamonds [55]. Only in rare cases, at low fraction of nitrogen, have the XEOL spectra for diamonds been registered [56]. Usually, the diamonds prepared by the HPHT method demonstrate a wide smooth band of X-ray exited luminescence (400–600 nm) without any peculiarities owing to nitrogen defects [42].

Since the total nitrogen content (C_N_ = 471 ppm, Table 3) exceeds the Eu amount by an order in magnitude, numerous nitrogen defects may influence the accommodation of Eu-atoms in diamonds. Indeed, our method provides cooperative doping diamonds with nitrogen and Eu. As we believed, these dopants mutually facilitate lattice modification. As far as both impurities intend to form complexes with vacancies, we did not exclude a preferable localization of Eu ions nearby nitrogen defects with vacancies. It is worth mentioning a formation of complexes of metal atoms with nitrogen (Ni-N, Co-N) in natural diamonds growing with dislocation-type defects in slip planes [57] and the incorporation of large impurity atoms into synthetic diamond (metal split-vacancy defects) [58]. There are some reasons to suppose a synergetic mechanism with the participation of nitrogen atoms which helps in the incorporation of heavy atoms into diamond lattice. Such a coordination could be partially retained during diamond synthesis when the amount of nitrogen atoms greatly exceeds the quantity of Eu atoms and their interactions with nitrogen atoms are very probable.

As we mentioned, the complexes of metal atoms with nitrogen (Ni-N, Co-N) were found in natural diamonds as a result of the growth of crystals with dislocation-type defects in slip planes [57]. Although in our case, the FTIR data did not reveal any complexing, some of nitrogen defects are able to decrease lattice stresses when heavy atoms are embedded into the diamond. The results of modeling [59,60] displayed a formation of B1-defects (4 nitrogen atoms associated with vacancy) which decrease the lattice energy comparative to perfect crystal. The effect is of (1–4) eV per one defect and depends on the amounts of nitrogen and hydrogen in crystal. Probably, a number of B1 defects surrounding the Eu atom in the lattice can compensate for the energy excess due to doping. However, in the IR spectrum (Figure 13) such defects are not revealed. On the whole, the role of nitrogen defects in the stabilization of Eu atoms in diamonds has not yet been elucidated.

Starting from the structure of pyrolyzate composed of carbon globules encapsulating Eu clusters, we do not deny their presence in synthesized diamonds with a doping degree much higher than that in synthetic or natural (carbonado) analogs [6,9,61]. The carbonado diamonds may contain the inclusions of hydrated rare-earth phosphates or metallic phases (Fe, Fe-Ni, Ni-Pt, Si, Ti, Sn, Ag, Cu) when the size of the inclusions is extended from few nanometers to much larger dimensions in the presence of aggregated nitrogen defects (N2V+, N2+, N3V, H1), indicating the synthesis of diamonds from a hydrocarbon material [9,61,62]. The studies of natural diamonds with rare earth elements and metallic phases indicate a principal possibility of artificial synthesis of such diamonds at high pressures and temperatures from hydrocarbons.

In our experiments, the HPHT process duration was chosen to fulfill a condition of the most complete precursor transformation into the diamonds when the precursor (pyrolyzate) is a conglomerate of nanosized globules captured europium atoms. Such globules have sufficient free volume for the formation of complexes of europium atoms with vacancies. In the crystallization process, the globules can incorporate also the carbon atoms released from pentaerythritol molecules.

However, according to our data, the release of europium atoms is inevitable during HPHT synthesis, and only tenths of a percent of the amount of europium in the precursor mixed with pentaerythritol are retained in diamonds. As a result of high temperature and pressure, europium is predominantly displaced from the diamond lattice and forms a hydroxide phase. In diamond crystals, a localization of europium in interstices is impossible due to the large size of the Eu atom. To insert an Eu^3+^ ion into the diamond lattice, about four carbon atoms must be removed. The inclusion of Eu clusters into the diamond lattice is even more impossible due to inadmissible huge crystal deformations.

The most energetically favorable option is the localization of individual europium atoms in the lattice sites with adjacent vacancies compensating the local stresses around the substitutional impurity. To clarify the subtle features of diamond modification with Eu, we simulated the growth of single crystals with defects including metal ion and vacancy, (Eu^3+^-V), and also built an ideal crystal (3d model, 528 atoms, the distance between the extreme atoms is ~1.8 nm) as a reference using the MM2 force field method [35,36].

Modeling [6] and our calculations of the formation of the (Eu^3+^-V) complex showed a moderate energy consumption, *E_Eu_* = 5.8 eV, being twice the binding energy of an atom in diamond (2.5 eV) [63]. We have simulated a formation of ideal nanocrystal and this one with Eu atom by sequential stacking of atoms in a growing structure. As a result, we found the positive energy difference (*E_Eu_*) between the data for defective and perfect crystals. 

It should be noted, the authors [6] simulated the formation of the (Eu^3+^-V) complex inside the lattice fragment (64 atoms) and determined the energy of the structure, *E_EuD_* ≈ 13.8 eV. Based on our modeling results, we evaluated the energy, *E_EuD_* ≈ 15.5 eV, for such small crystal fragment by eliminating the contribution of external carbon atoms around this fragment. Both quantities are comparable, although the simulation procedures were different and an exact match cannot be expected.

Finally, taking the data (Table 1) for the samples (SEu2, SEu3) with moderate and high content of diamonds, we calculated the Eu amount in the limit of pure diamond, *C*_EuD_ = 51 ± 5 ppm. In addition, we also determined the Eu content in graphite, *C*_EuG_ = 128 ± 5 ppm, that is twice and a half greater than the Eu content in the diamond lattice.

In our synthesis, to saturate the diamonds with Eu up to the concentration *C*_EuD,_ it takes relatively small molar energy cost, *E*_Eu_*C*_EuD_~2.4 J/g, which corresponds to the sum of kinetic energies of ~30 carbons per Eu defect at the temperature *T*~2100 K in the HPHT process when a carbon atom gets the energy *E*_1_~*k*_B_*T*~0.2 eV. If we imagine, hypothetically, a transformation of a globule with Eu cluster (~3 atoms) in pyrolyzate into a diamond particle, the total kinetic energy of carbons (~90) in the shell around the cluster is enough to cover the energy cost for the rearrangement of carbons into the diamond lattice. However, a formation of metallic clusters in the lattice during HPHT synthesis seems to be problematic due to Eu atom displacements by the moving growing crystalline boundaries. As a result, the diamond phase has kept only several tenths of a percent of the initial amount of Eu in the precursor. On the other hand, such a cluster in diamond lattice should induce huge structural distortions and local stresses. Energetically, a distribution of single Eu atoms substituting carbons and stabilized by vacancies in the neighborhood seems more profitable. Until now, a preferable mode of integration of lanthanides into the diamond lattice remains an open question in theory and experiment [64,65,66,67].

## 4. Conclusions

We have developed a new method of micrometer sized diamond synthesis by using a heat-resistant ultraporous precursor being diphthalocyanine pyrolyzate. As a result, we have obtained the diamonds doped with Eu atoms up to the concentrations *C*_EuD_ = 51 ± 5 ppm, almost two orders of magnitude higher than those in natural diamonds and synthetic analogues. The improved precise technique of X-ray fluorescence measurements allowed us to determine, reliably and precisely, the Eu contents in the samples upon their activation by irradiation.

The incorporation of Eu atoms into the diamond lattice was facilitated by a metal–carbon precursor composed of multilayered globular nanostructures with small clusters of heavy atoms inside carbon shells which, to a certain extent, retained them during HPHT synthesis. A transformation of such globules into diamond structures has led to the formation of Eu defects where heavy atoms substitute carbons in the lattice. Its distortion was compensated by adjacent vacancies that were modelled by the MM2 force field method applied to describe the growth of a diamond nanocrystal with a single Eu^3+^ ion. A simulation has allowed us to calculate the energy cost of 5.8 eV for defect formation at the level of twice the binding energy of atoms in diamond.

These results are consistent with the XEOL measurements showing a characteristic luminescence from Eu^3+^ ions in diamonds exposed by X-rays. The relationship between the intensities of characteristic lines of luminescence corresponding to magnetic dipole transition (^5^D_0_–^7^F_1_) and hypersensitive forced electric dipole transition ^5^D_0_ → ^7^F_2_ has indicated the absence of centrosymmetric positions of Eu^3+^ ions in the diamond lattice due to vacancies coupled with ions.

In the diamond lattice, we detected also a large amount of nitrogen by one order of magnitude higher than heavy atom content. Since the nitrogen impurities often form the defects containing vacancies, the nitrogen may serve as a component intensifying the incorporation of Eu atoms by the interactions between both type defects.

Assuming relevant applications of micrometer sized diamonds with highly luminescent dopants, we consider the developed methodology to be a promising tool for the development of the diamond doping industry using special type nano-precursors.

## Figures and Tables

**Figure 1 materials-16-00830-f001:**
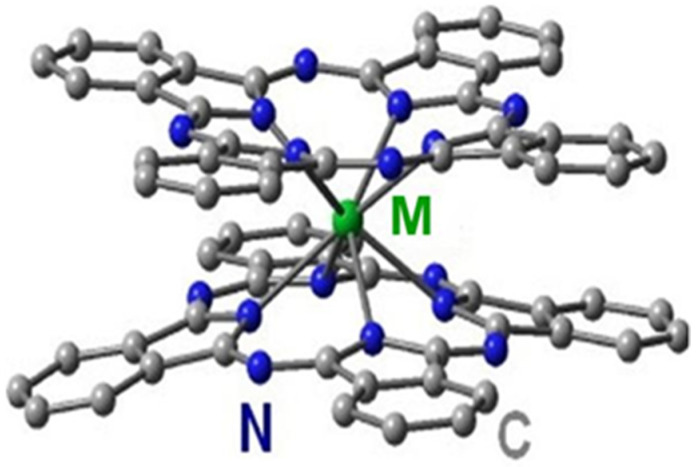
Diphthalocyanine molecule with a metal atom (M = Eu) linking plane organic ligands via nitrogen atoms.

**Figure 2 materials-16-00830-f002:**
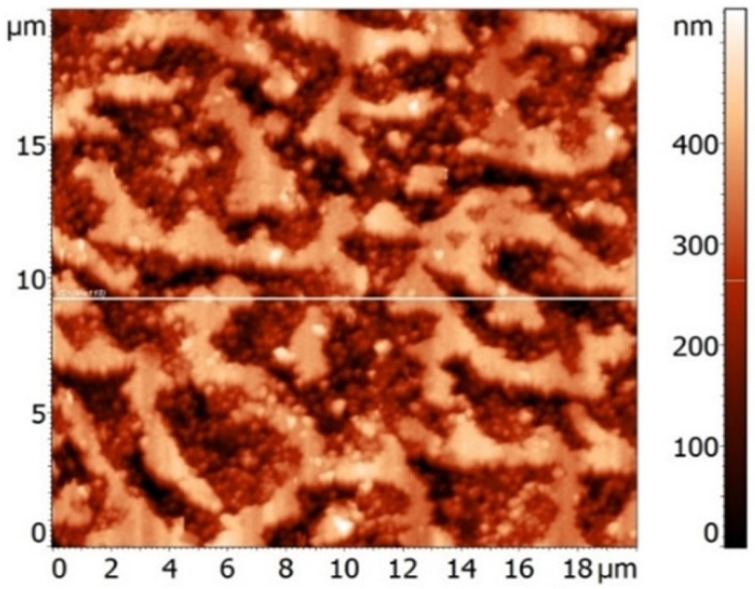
AFM image of the surface of pyrolyzate composed of nanoscale and submicron aggregates forming a porous matrix.

**Figure 3 materials-16-00830-f003:**
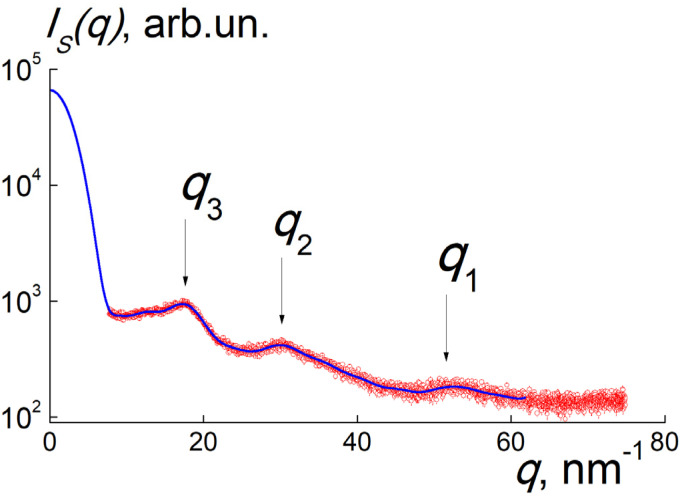
Intensity of X-ray scattering *I*(*q*) on pyrolyzate (red dots) vs. scattering vector modulus. The positions of peaks (*q*_1,2,3_) are marked. Solid curve (blue) corresponds to the data approximation using correlation spectrum *G*(*R*).

**Figure 4 materials-16-00830-f004:**
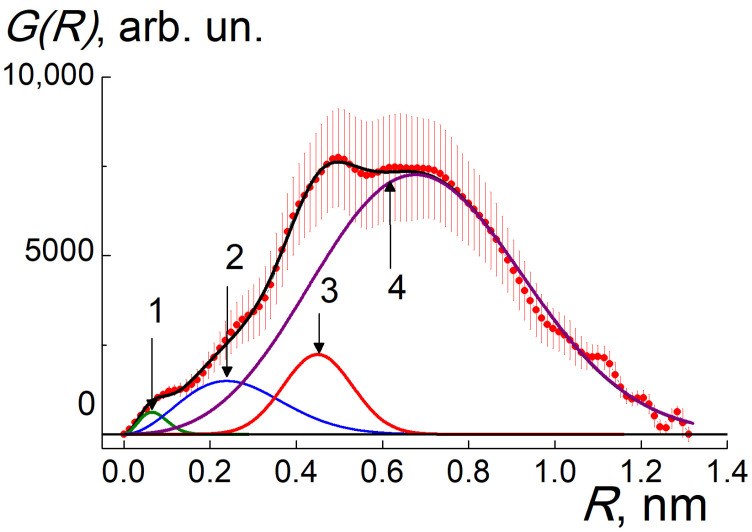
Distances distribution *G*(*R*) (dots) between scattering centers in pyrolyzate (carbon atoms, Eu-clusters, fragments of carbon shell around them) fitted with the function (1) (black line). Partial correlation functions 1–4 (lines) are plotted: 1—green, 2—blue, 3—red, 4—violet.

**Figure 5 materials-16-00830-f005:**
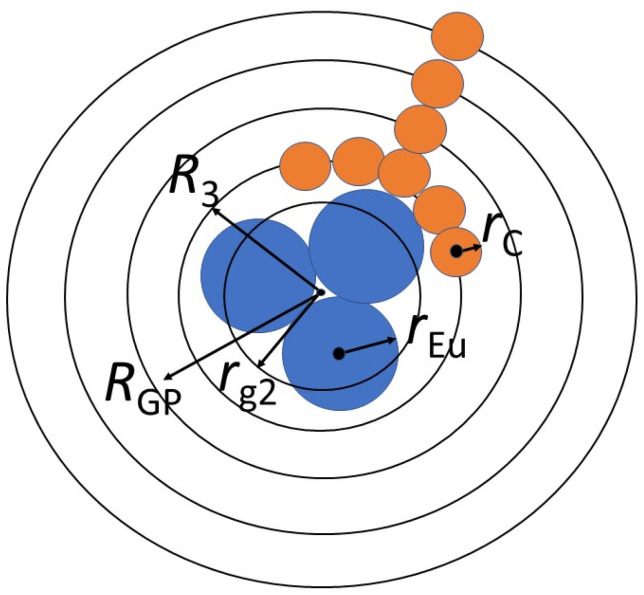
Spherical model of globule with central Eu-cluster and carbon shell (4 layers): *r*_C_ and *r*_Eu_ are atomic radii for carbon and Eu; *r*_g2_ and *R*_GP_ are gyration radii for Eu-cluster and globule; *R*_3_ denotes a location of the carbon layer contacting with the Eu-cluster.

**Figure 6 materials-16-00830-f006:**
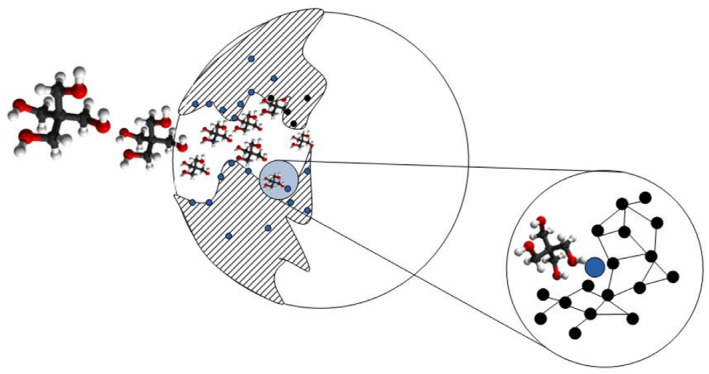
Scheme of filling pyrolyzate with pentaerythritol molecules composed of carbon (black), oxygen (red) and hydrogen (white) atoms. The molecules penetrate into the pores of the matrix of carbon (black) and interact with released Eu atoms (blue) at pores border by diamond synthesis. In carbon matrix material the Eu inclusions (blue dots) are shown also.

**Figure 7 materials-16-00830-f007:**
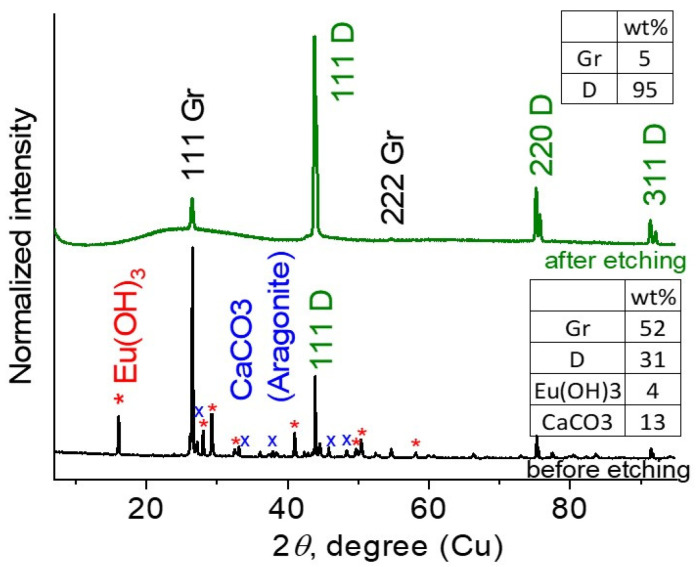
XRD patterns for the samples SEu1, Seu3. D—diamond, Gr —graphite. Reflexes from different phases are denoted. The “*” and “x” symbols are related to the reflects from Eu(OH)_3_ and CaCO_3_ phases respectively. The weights of phases are shown in tables.

**Figure 8 materials-16-00830-f008:**
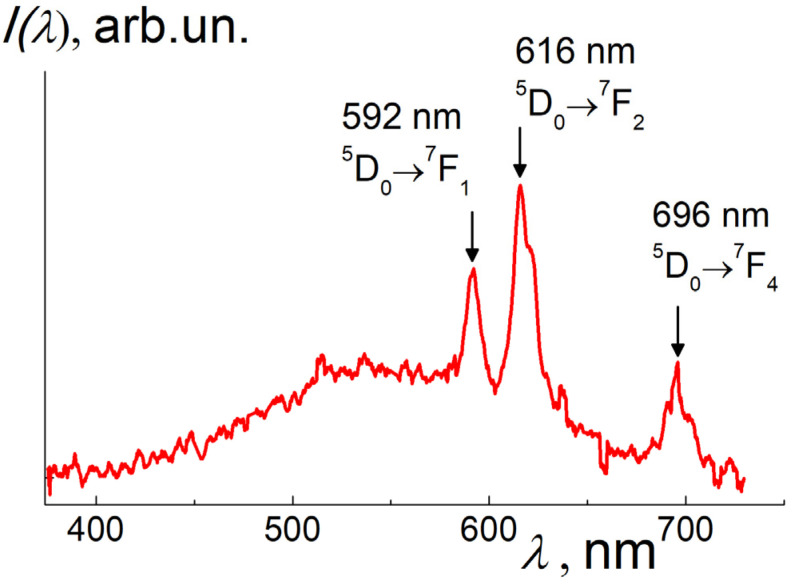
XEOL spectrum (excitation energy 8 keV) for the powder sample SEu1. The Eu-peaks at characteristic wavelengths and corresponding electron transitions are marked. The luminescence from the phases of graphite, diamonds, and aragonite is visible in a wide band (400–700 nm).

**Figure 9 materials-16-00830-f009:**
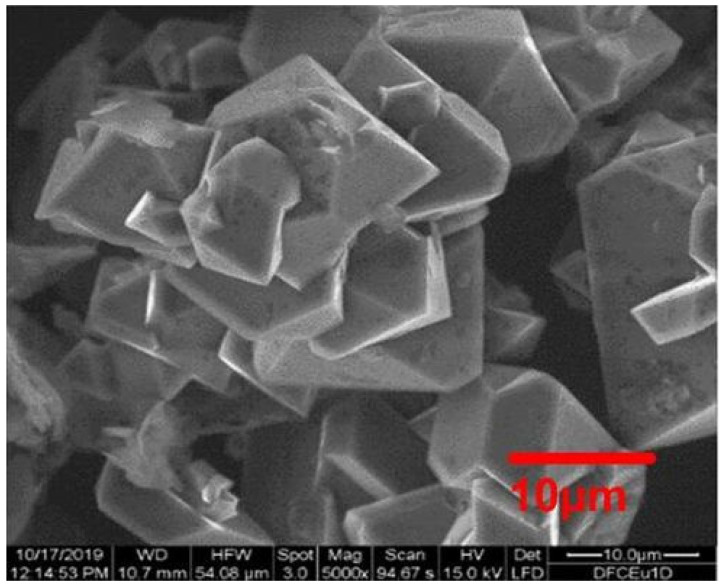
SEM image of purified diamond powder SEu3. Micrometer sized diamond crystals with sharp borders and plane facets are visible.

**Figure 10 materials-16-00830-f010:**
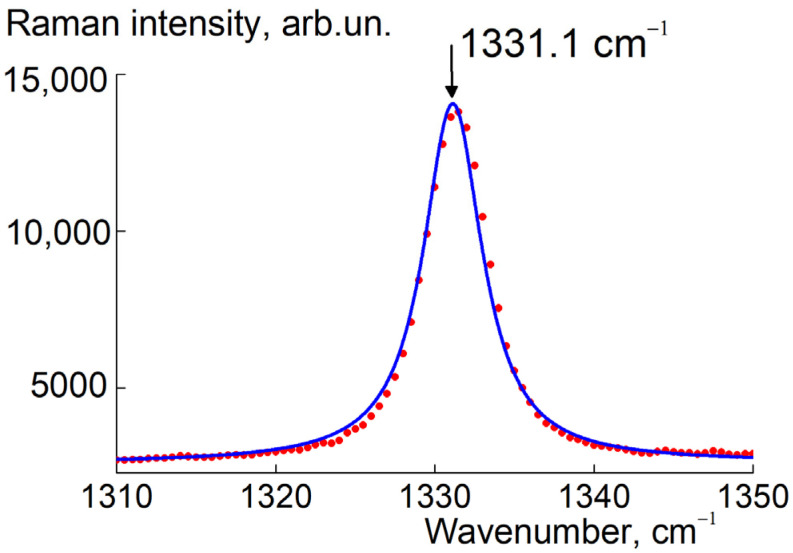
Raman spectrum for the purified sample SEu3 (dots). Lorentzian fitting (solid line) shows a characteristic diamond peak with maximum at 1331.1 cm^−1^.

**Figure 11 materials-16-00830-f011:**
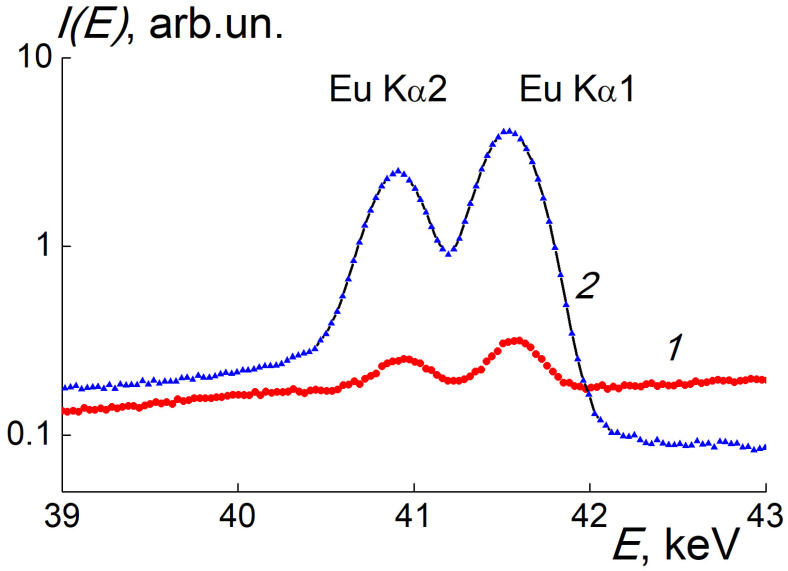
XRF@241Am spectra for the SEu3 (1) and the reference sample (2). The lines Eu Kα2 and Eu Kα1 are marked. Comparison of peaks amplitudes for the probes (1,2) allowed us to find the Eu concentration in the SEu3 sample enriched with diamonds.

**Figure 12 materials-16-00830-f012:**
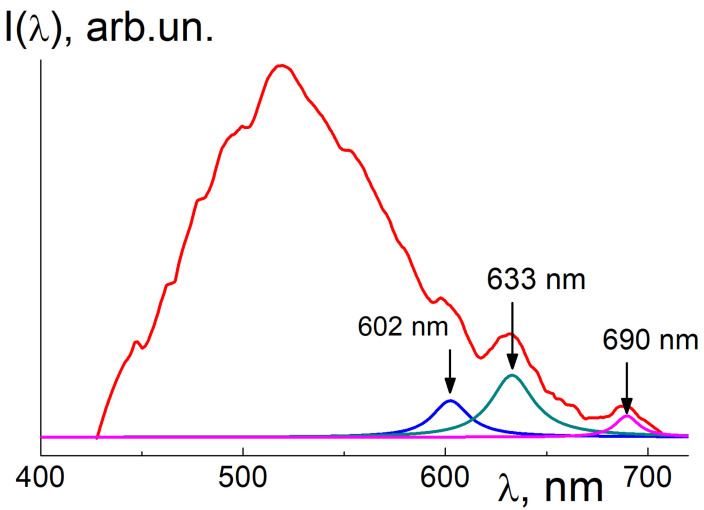
XEOL spectrum for the SEu3 sample (red curve) showing Eu^3+^ characteristic bands corresponding to the electron transitions ^5^D0 →^7^F_1,2,4_ (maxima at 602; 633; 690 nm). Lorentzian fitting (blue, green, violet curves) for the bands is shown with maxima indications.

**Figure 13 materials-16-00830-f013:**
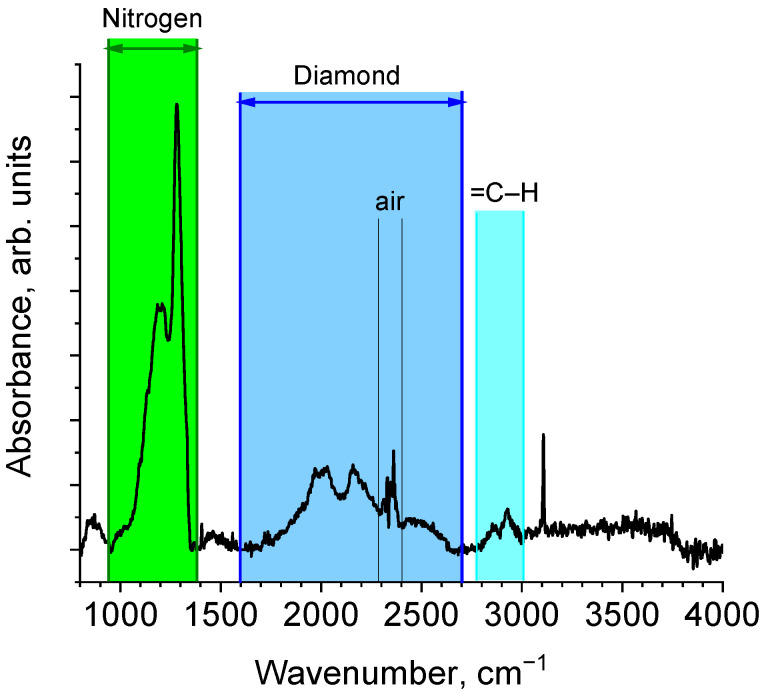
IR spectrum for the sample SEu3. The ranges of wavenumbers related to various nitrogen defects (Table 3) and other absorption centers in diamonds are shown.

**Table 1 materials-16-00830-t001:** Diamond containing samples with Eu.

Sample	Eu Content	Purification, HCl	Treatment, CHBr_3_	Diamond, wt.%	Graphite, wt.%
SEu1	5.7 wt.%	–	–	–	–
SEu2	88 ± 5 ppm	+	–	52	48
SEu3	55 ± 5 ppm	+	+	95	5

**Table 2 materials-16-00830-t002:** Correlation lengths and dispersions in Equation (1).

*r*_1_, nm	*r*_2_, nm	*R*_3_, nm	*r*_3_, nm	*R*_4_, nm	*r*_4_, nm
0.065 ± 0.003	0.24 ± 0.03	0.42 ± 0.002	0.11 ± 0.03	0.44 ± 0.05	0.40 ± 0.02

**Table 3 materials-16-00830-t003:** Concentration of nitrogen defects according to FTIR data.

Wavenumber, cm^−1^	Nitrogen Defect Type	Concentration, ppm
1132	N^0^ (C or P1)	156
1280	2N (A-aggregate)	287
1331	N^+^ (C^+^)	28
Sum		471

## Data Availability

No new data were created.

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
