# Peer review of "X-ray Excited Optical Luminescence of Eu in Diamond Crystals Synthesized at High Pressure High Temperature"

_materials, 2023, doi:10.3390/ma16020830_

Round 1
Reviewer 1 Report
The manuscript is interestng, and well written.
It contains many technicalities, not easily understood for a non epert as I am;
anyway the overall presentation onthese parts is quite convincing
Author Response
Dear Referee,
Authotrs are very grateful for the reviewing and agree with the remarks.
In revised version we checked english and style corrected for better quality of manucript.
Reviewer 2 Report
Comments are in the attachment.

Author Response
Authors thank Rereree, comments are enclosed.

Reviewer 3 Report
The study presented in this research is sound, and the results produced are interesting. But a revision is required, and after responding to the following remarks and revising the paper, the manuscript may be considered for publication.
1. Literature review needs to include several recent, relevant publications (high impact) highlighting their key findings. The current version only discussed general aspects while the review of each from several papers is necessary. You may provide a review summary table consisting of a column for the comments or key conclusions.
2. More recent relevant literature or similar work discussion is mandatory in the introduction section, which is missing in the Introduction. Authors are suggested to add one paragraph in the introduction section by discussing the recent progress and citing similar work.
3. The novelty of the work is missing in the introduction. Authors are suggested to include a separate paragraph discussing the novelty and importance of the present work.
4. Authors are suggested to include a literature review on the recent publication on rare-earth doping and its applications based on the following references in the introduction section: DOIs: 10.1021/acsaelm.1c00703; 10.1021/acsaelm.2c00069;
10.1016/j.ceramint.2022.07.220..
5. Also, check the typos throughout the manuscript during revision submission.
Author Response
Authors thank Referee, comments are enclosed

Round 2
Reviewer 2 Report
Thanks to the authors for the effort their made to improve the paper. The paper is of interest and in the corrected version should be well understood by potential readers.